# Racemate Resolution of Alanine and Leucine on Homochiral Quartz, and Its Alteration by Strong Radiation Damage

**DOI:** 10.3390/life11111222

**Published:** 2021-11-11

**Authors:** Adrien D. Garcia, Cornelia Meinert, Friedrich Finger, Uwe J. Meierhenrich, Ewald Hejl

**Affiliations:** 1Institut de Chimie de Nice, Université Côte d’Azur, CNRS, UMR 7272, 06108 Nice, France; adrien.garcia@univ-cotedazur.fr (A.D.G.); uwe.meierhenrich@univ-cotedazur.fr (U.J.M.); 2Fachbereich Chemie und Physik der Materialien, Universität Salzburg, Hellbrunnerstraße 34, 5020 Salzburg, Austria; friedrich.finger@sbg.ac.at; 3Fachbereich für Geographie und Geologie, Universität Salzburg, Hellbrunnerstraße 34, 5020 Salzburg, Austria

**Keywords:** chirogenesis, quartz, amino acids, radiation damage, origin-of-life, GC×GC-TOFMS

## Abstract

Homochiral proteins orchestrate biological functions throughout all domains of life, but the origin of the uniform l-stereochemistry of amino acids remains unknown. Here, we describe enantioselective adsorption experiments of racemic alanine and leucine onto homochiral *d*- and *l*-quartz as a possible mechanism for the abiotic emergence of biological homochirality. Substantial racemate resolution with enantiomeric excesses of up to 55% are demonstrated to potentially occur in interstitial pores, along grain boundaries or small fractures in local quartz-bearing environments. Our previous hypothesis on the enhanced enantioselectivity due to uranium-induced fission tracks could not be validated. Such capillary tubes in the near-surface structure of quartz have been proposed to increase the overall chromatographic separation of enantiomers, but no systematic positive correlation of accumulated radiation damage and enantioselective adsorption was observed in this study. In general, the natural *l*-quartz showed stronger enantioselective adsorption affinities than synthetic *d*-quartz without any significant trend in amino acid selectivity. Moreover, the l-enantiomer of both investigated amino acids alanine and leucine was preferably adsorbed regardless of the handedness of the enantiomorphic quartz sand. This lack of mirror symmetry breaking is probably due to the different crystal habitus of the synthetic z-bar of *d*-quartz and the natural mountain crystals of *l*-quartz used in our experiments.

## 1. Introduction

Homochiral polymers must have emerged rather early during abiogenesis, most probably in a pre-RNA/DNA/metabolism world [1,2,3]. Because of enantiomeric cross-inhibition, the synthesis of polypeptides and the self-replication of polynucleotides is repressed or even impossible in a racemic environment [4,5,6]. Consequently, homochirality or at least a substantial enantiomeric enrichment (*ee*) in the chemical building blocks of the first functional biopolymers should have developed early during chemical evolution. While the chronological details are still unclear, it is well accepted that chemical evolution did not occur suddenly; instead, a gradual transition from simple organic building blocks, including chiral molecules toward polymeric and supramolecular assemblies, has built the fundaments for self-assembling, self-sustaining interactive systems with emerging patterns that have ultimately evolved into what could be considered as living entities [7,8].

Abiotic synthesis of asymmetric molecules, such as amino acids or sugars, usually produces racemic compositions. Several mechanisms capable to entail molecular symmetry breaking leading to biomolecular homochirality on Earth, so-called chirogenesis, have been proposed and investigated. The most popular hypotheses comprise symmetry breaking by crystallization [9], selective adsorption of enantiomers on mineral surfaces [10,11,12], molecular parity violation [13], and enantioselective chemistries induced by circularly polarized light [14,15,16,17,18] or spin-polarized electrons [19].

Regarding terrestrial chirogenesis due to enantioselective adsorption on minerals, the putative role of naturally etched nuclear particle tracks, for example fission tracks under early Archean weathering conditions, has not yet been evaluated experimentally. Such capillary tubes in enantiomorphic minerals can eventually produce a local chiral enrichment of amino acids or other pre-existing biomolecules. Hejl has proposed a hypothesis for racemate resolution by chemical etching of nuclear particle tracks, as for example fission tracks originating from the spontaneous or induced fission of ^238^U or ^235^U, respectively [20]. The hypothesis relies on the principle of liquid chromatography, i.e., on the chemical separation of a mixture of dissolved chemical components (*mobile phase*) passing through a capillary structure (*stationary phase*). Molecular separation is supposed to occur in etched fission tracks, serving as a *stationary phase* with respect to a mobile solution of molecules. According to this hypothesis, fission tracks in enantiomorphic minerals could produce chiral separation of a racemic mixture, in addition to the chiral separation on their external crystal faces, cleavage planes, and fractures. In other words, accumulated radiation damage by nuclear particle tracks could increase the overall chiral resolution effect of enantiomorphic minerals—especially when they are chirally enriched on a local scale. Despite its low uranium content (<1 µg g^−1^), natural occurring quartz may contain clusters of fission tracks around uranium-rich inclusions. Such track clusters in quartz were used in the past for fission-track dating [21,22,23]. Adequate etching conditions for fission tracks in quartz have already been published by Fleischer and Price [24]. Note, however, that quartz does not incorporate uranium in its crystal lattice.

Besides quartz, most enantiomorphic mineral species are either rare or have probably not occurred on the early Earth. With regard to eventual chiral enrichment by autocatalytic secondary nucleation, Hejl and Finger have investigated chiral proportions of nepheline in rocks undersaturated in SiO_2_ which had crystallized from melts with low viscosity [25]. They found that nepheline mainly occurs in racemic proportions, even in small volumes of rock. Furthermore, nepheline-bearing rocks are not known from Archean terrains. They first appeared on Earth about 2.7 Ga ago [26], i.e., about 1 Ga later than the eldest traces of life. Enantiomorphic zeolites were considered for racemate resolution [27], but their ubiquitous presence on the early Earth is doubtful. Other enantiomorphic mineral species, as for example berlinite, pinnoite, wardite, or leucophanite are extremely rare and not known from the early Archean crust. 

Therefore, quartz is a more promising mineral for terrestrial chirogenesis to have potentially generated racemate resolution. The presence of a granitic crust and, thus, the widespread occurrence of quartz on the early Earth are testified by detrital zircons in gneisses of the Yilgarn craton in Western Australia. The oldest ever zircon has a U-Pb age of 4.404 ± 0.008 Ga [28]. Chemical zoning and oxygen isotope composition of this zircon indicate that it has crystallized from a granitic magma that had been molten from supracrustal rocks in contact with an early hydrosphere [28]. 

Bonner et al. have first demonstrated asymmetric adsorption of alanine on crystal faces of quartz [11], but it was not considered as an effective process for regional or global racemate resolution because *d*- and *l*-quartz occur in equal proportions on a global scale [29,30,31]. However, sizeable volumes of homochiral quartz were reported from hydrothermal veins [32], and from graphic granite [33]. Thus, local enrichments of either *d*- or *l*-quartz on the early Earth may have occurred.

Even non-enantiomorphic minerals can exhibit surfaces with a chiral atomic pattern [34,35,36], but such chiral crystal faces occur pairwise with opposite handedness on the same crystal. Thus, crystal faces with opposite handedness are equally frequent in granulates of a non-enantiomorphic mineral species. Any chiral separation effect on crystal faces or cleavage planes will be compensated in a relatively small volume of rock, granulated sediment, or soil. On the other hand, chiral faces do not occur pairwise with opposite handedness on a single enantiomorphic crystal. Its whole surface has a biased chiral pattern, and a granulated enantiomorphic mineral with similar handedness (either *d* or *l*) may have an enantioselective separation effect on a percolating racemic solution of chiral molecules.

Here, we experimentally demonstrate significant racemate resolution in percolating solutions of racemic alanine and leucine on homochiral quartz sand with randomly orientated surfaces (i.e., without well-defined crystallographic orientation). Moreover, our results partly disprove the amplifying role of etched nuclear particle tracks according to the hypothesis of Hejl on the enantioselective adsorption potential of enantiomorphic quartz [20]. 

## 2. Materials and Methods

### 2.1. Characterization and Preparation of Homochiral Quartz Samples

Homochiral quartz sand was produced in two different ways. A first sample was obtained by crushing and sieving a synthetic z-bar quartz from a commercial provider (Hausen GmbH, Telfs, Austria). Such quartz crystals grow under hydrothermal conditions on so-called seed plates that are produced by cutting already existing z-bar crystals in thin plates of well-defined crystallographic orientation. A commercial laboratory will, thus, always duplicate z-bars of the same handedness. They are not twinned and strictly homochiral *l*- or *d*-quartz. Our z-bar quartz crystal had a mass of 1473 g (Figure 1a). Prior to crushing and sieving, we determined its sense of optical rotation for light propagation parallel to the crystallographic c-axis. For this purpose, we produced a quartz plate by cutting the crystal with a diamond saw perpendicular to the crystallographic c-axis. The plate was ground with silicon carbide, and polished with diamond paste down to a grain size of 1 µm. Its thickness of 3.16 mm was measured with a vernier calliper with a precision of ±0.01 mm. The sense and angle of optical rotation of the transparent plate were determined with an optical polarimeter (Modular Compact Polarimeter MCP 150, Anton Paar GmbH, Graz, Austria). The temperature of the sample during the measurement was kept constant using Peltier elements. Clockwise optical rotation corresponds to *d*-quartz, and counter-clockwise optical rotation corresponds to *l*-quartz when rotation is observed in the sense opposite to the light propagation [37,38]. Our z-bar crystal was found to be *d*-quartz with an optical rotatory power of +21.68 ° mm^−1^ at 20 °C. 

The pieces of z-bar quartz were treated by alternating crushing and sieving, in order to produce sand with a grain size of 0.25–0.50 mm. The quartz pieces were first broken with a hammer on a steel plate, and then shortly crumbled with a disk mill for 5 s only. After each sieving step, the coarser fraction (>0.50 mm) was crushed and sieved again. Finally, we obtained about 400 g of homochiral *d*-quartz sand.

This quartz sand was washed with acetone, isopropanol, and double-distilled water, and dried in a drying chamber at 105 °C. The dried sand was purified with a Frantz magnetic separator to remove small steel particles that may have been produced by abrasion of the steel hammer or the mill. Afterwards, this purified sand was heated for 3 h at 500 °C in a muffle furnace to minimize biological contaminations. An aliquot of the glowed sand was then powdered in an agate mill and chemically analyzed by X-ray fluorescence, revealing the composition to be of 99.88% SiO_2_ with minor contaminations of Fe and Cr, probably from abrasion of the steel hammer or the mill.

Unfortunately, we could not purchase synthetic *l*-quartz due to the method of production of z-bar quartz. We found only z-bar providers which offer either *d*-quartz or give no specification on the handedness of the quartz. Therefore, we decided to produce *l*-quartz sand by crushing and sieving crystallographically well-defined mountain crystals, which had grown under natural conditions. First, we evaluated 20 natural quartz crystals from an open tension joint in Madagascar (Figure 1b). These transparent crystals of fairly similar size and habitus had a total mass of 507 g with individual masses between 13.7 and 40.1 g, and individual lengths ranging from 5 to 9 cm. The longitudinal axes were found to be the crystallographic c-axis. Ten of these crystals were rejected because they were either twinned or intergrown with smaller quartz crystals. The remaining ten crystals were found to be single crystals without twinning. They had a total mass of 227 g and were chosen for further treatment.

These single crystals were cut with a diamond saw perpendicular to their c-axis, to produce plates with a thickness of about 3 mm (Figure 2). These plates were ground with silicon carbide and polished with diamond paste, in a similar way as for the plate from the z-bar. Six crystals were found to be *l*-quartz with optical rotatory powers between 21.26 and −22.74° mm^−1^ at 20 °C (Table 1). The small variations of the optical rotatory power are probably due to a slight imprecision of the cutting angle relative to the c-axis (ca. 90° ± 2°). The other four crystals were found to be *d*-quartz with a positive optical rotatory power and, therefore, rejected. The material from the six *l*-quartz crystals was treated by alternating crushing and sieving, washed, purified, and glowed in a similar way as the z-bar sample. Finally, we obtained 44 g of homochiral *l*-quartz sand with a grain size of 0.25–0.50 mm. The specific surface area (SSA) of both kinds of quartz sand (*l* and *d*) can be deduced from their quasi equigranular grain size (0.25–0.5 mm), using published data of granulometric investigations [39]. Recovered amino acid quantities normalized to the specific surface area were calculated with a SSA of 60 m^2^ g^−1^ and a porosity of 40% that is typical for equigranular sand with angular grains.

### 2.2. Thermal Neutron Irradiation of Homochiral Quartz Sand Mixed with Uraniferous Zircon Powder

Synthetic z-bar quartz nor mountain quartz contain any uraniferous inclusions. In order to produce fission tracks on the fractured surfaces of quartz grains, aliquots of the homochiral sand samples were mixed with uraniferous zircon powder. For this purpose, zircon sand from a placer deposit in New South Wales (Australia) was ground to fine powder in an agate mill. The chemical composition of the zircon powder was analyzed by X-ray fluorescence using a Bruker Pioneer S4 crystal spectrometer at the Department for Chemistry and Physics of Materials (University of Salzburg). Obtained net count rates on single X-ray lines were recast into concentration values (wt% and ppm) based on an in-house calibration routine that involves measurements of ca. 30 international geo-standards (USGS and GSJ). The calibration was based on the Bruker AXS software SPECTRAplus FQUANT (v1.7) which corrects absorption, fluorescence, and line overlap effects. Concentrations of U and Th were 0.049 and 0.018 wt%, respectively (Table 2). 

The zircon powder was glowed in a muffle furnace at 500 °C for 3 h to reduce the risk of biological contaminants, such as natural l-amino acids. Afterwards, it was mixed with aliquots of either homochiral *d*- or *l*-quartz sand. Thus, 125 g of *d*-quartz sand were mixed with 25 g of zircon powder, and 12.5 g of *l*-quartz sand were mixed with 2.5 g of zircon powder corresponding to an identical mass ratio of 88.33% quartz and 16.67% zircon. Both mixtures were filled in polyethylene tubes. When the surface of quartz grains is completely covered by zircon powder with a uranium concentration of 0.049 wt%, a thermal neutron fluence of 5 × 10^15^ neutrons cm^−2^ will induce a fission-track density of about 10^7^ tracks per cm^2^ onto the quartz surface [40]. The addition of zircon powder alone would not lead to enough fission tracks within an adequate time scale as the spontaneous fission of ^238^U is a very rare event. It would take several million years to produce a significant quantity of fission tracks on the surface of quartz. We, thus, induced the fission of ^235^U by subsequent neutron irradiation of the mixed quartz sand/zircon powder.

Thermal neutron irradiation of the filled tubes was performed in the TRIGA Mark-II Reactor of the TU Vienna (Atominstitut der TU Vienna, Austria). This research reactor of the swimming pool type has a nominal power of 250 kW. The reactor is controlled by three control rods which contain boron carbide as an absorber material. The thermal neutron flow rate is in the order of 10^13^ cm^−2^ s^−1^ in a central position. Desired neutron fluences of about 5 × 10^15^ neutrons cm^−2^ were obtained after an irradiation time of about 1 h, with irradiation position at the border of the reactor core. After a decay time of two months, the capsules possessed total activities of up to 200 kBc—mainly due to the activities of ^46^Sc, ^95^Zr, ^95^Nb, ^181^Hf, and ^175^Hf. Besides uranium fission tracks, the samples must have accumulated a rather high radiation damage by beta decay and gamma radiation.

### 2.3. Sand Etching and Subsequent Infiltration with Aqueous Solutions of dl-alanine and dl-Leucine

Irradiated *d*- and *l*-quartz sand, as well as nonirradiated *d*- and *l*-quartz, were etched with 10% HF. This etching transforms superficial latent fission tracks into capillary tubes [24,40]. Identical etching conditions were applied to the nonirradiated *d*- and *l*-quartz samples to avoid any differences in the adsorption properties caused by radiation damage vs. etching, allowing for a better comparison of the enantioselective adsorption with the irradiated samples.

The sand fractions were filled in plastic Erlenmeyer flasks together with 10% HF and were etched at 20 °C for 2 h under permanent agitation on a shaking table. Afterwards, the sand was rinsed several times with double-distilled water. A second etching with 10% HCl was applied for 20 min, to dissolve potential fluorine components. While repeatedly rinsing the etched sand with double-distilled water, the zircon powder was removed by decantation. Finally, the sand was desiccated at 95 °C in a drying chamber. 

Subsequent adsorption experiments of both *d*- (synthetic) and *l*-quartz (natural), respectively, with diluted aqueous solutions of individual racemic amino acids (dl-Ala and dl-Leu) were conducted as follows. The glassware (Erlenmeyer flasks, beakers, watch glasses, and volumetric flasks) as well as needles, spatulas, and aluminum foil were cleaned with acetone and isopropanol, and then glowed for 3 h in a muffle furnace at 500 °C. Solutions of racemic amino acids with concentrations of 10^−3^ M were prepared in volumetric flasks. This amino acid concentration was chosen to allow for accurate quantitation of recovered amino acids. Glass funnels were placed on the Erlenmeyer flasks, and conically folded aluminum foils were placed in the funnels. These foils were checked for impermeability prior to utilization. First, 4 mL of 10^−3^ M racemic amino acid solutions (either dl-Ala or dl-Leu) were dropwise added to the funnels, followed by homochiral quartz sand (either *d*-quartz or *l*-quartz) up to the liquid surface. This procedure avoided the formation of air bubbles, and the interstices between the sand grains became entirely filled with the racemic solutions. Eight funnels were prepared in this way, i.e., dl-Ala + nonirradiated *d*-quartz (1), dl-Ala + irradiated *d*-quartz (2), dl-Leu + nonirradiated *d*-quartz (3), dl-Leu + irradiated *d*-quartz (4), dl-Ala + nonirradiated *l*-quartz (5), dl-Ala + irradiated *l*-quartz (6), dl-Leu + nonirradiated *l*-quartz (7), and dl-Leu + irradiated *l*-quartz (8). In addition, four blanks with nonirradiated *d*- and *l*-quartz as well as irradiated *d*- and *l*-quartz were filled with 4 mL of double-distilled water. 

The funnels filled with dl-amino acids and homochiral sand were covered with watch glasses and were allowed to stand for 72 h at 20 °C. Afterwards, a clean needle—with a thickness smaller than the grain size of the sand—was inserted vertically in the sand body and pushed downward to the bottom of the foil to perforate it. Then, 8 mL of double-distilled water was dropwise added to the top of the sand. In this way, the aqueous solution containing the diluted amino acid depleted in one adsorbed enantiomer was rinsed out from the pores between the sand grains and recovered in a small beaker and then filled in vials with silicon screw caps.

In a previous experiment, thin glass tubes were used instead of funnels. The lower end of these tubes was closed with aluminum foil and silicone. Mixed solutions containing 10^−3^ M dl-Ala and 10^−3^ M dl-Leu were used in the following manner: dl-Ala + dl-Leu + nonirradiated *d*-quartz (9) and dl-Ala + dl-Leu + irradiated *d*-quartz (10). In addition, two blanks with nonirradiated and irradiated *d*-quartz, both with 4 mL of double-distilled water, were prepared. 

### 2.4. GC×GC-TOFMS Analysis—Sample Preparation and Derivatization

All sample-handling glassware was wrapped in aluminum foil and heated at 500 °C for 3 h prior to usage. Eppendorf tips^®^ were sterile, and the water used for standard solutions, reagent solutions, and blanks was prepared with a Direct-Q^®^ water purification system (<3 ppb total organic carbon). All solvents, reagents, and amino acid standards were purchased from Sigma-Aldrich and stored according to the respective instructions.

Specific volumes of the eluate of each infiltration experiment described above were transferred into a 1 mL of Reacti-Vial^TM^ and dried under a gentle flow of nitrogen. These volumes were chosen to reach final concentrations that resulted in adequate peak intensities and areas for reliable enantiomeric excess quantification. The chosen volumes were as follows: 75 µL of (1) dl-Ala + nonirradiated *d*-quartz, 100 µL of (2) dl-Ala + irradiated *d*-quartz, 75 µL of (3) dl-Leu + nonirradiated *d*-quartz, 100 µL of (4) dl-Leu + irradiated *d*-quartz, 75 µL of (5) dl-Ala + nonirradiated *l*-quartz, 200 µL of (6) dl-Ala + irradiated *l*-quartz, 75 µL of (7) dl-Leu + nonirradiated *l*-quartz, 200 µL of (8) dl-Leu + irradiated *l*-quartz, 50 µL of (9) dl-Ala + dl-Leu + nonirradiated *d*-quartz, and 50 µL of (10) dl-Ala + dl-Leu + irradiated *d*-quartz.

All samples were re-dissolved in 50 µL of 0.1 M HCl, and individually derivatized to form *N*-ethylchloroformate heptafluorobutyryl esters (ECHFBE) by adding first 6 µL of pyridine and 19 µL of heptafluorobutanol followed by 5 µL of ethylchloroformate. The reaction mixtures were vortexed for 10 s, and the derivatives were extracted and transferred into GC vials by adding 50 µL of chloroform containing methyl myristate as an internal standard at a concentration of 10^−6^ M. Three aliquots of each incubated quartz sample (1) to (10) were derivatized, and each derivatized sample was analyzed (1 µL) as triplicates by enantioselective two-dimensional gas chromatography coupled to a time-of-flight mass spectrometer GC×GC-TOFMS [41,42], i.e., each quartz sample was analyzed nine times by preparing three derivatized aliquots with *n* = 3 to obtain sufficient replicates for the statistical analysis.

Racemic standard solutions of alanine and leucine—identical to the standards used for the adsorption experiments—with approximately the same concentration than the final quartz samples were used to determine the enantiomeric bias of the commercially available racemic standards. These reference standards were injected three times in between each triplicate injections of the three aliquots of the same quartz infiltration experiment to follow any chromatographic bias over time. In total, nine injections of the alanine and leucine reference standard, respectively, were performed for one adsorption sample, i.e., three individually derivatized reference standards with *n* = 3 injections. To ensure that no contamination occurred during the analytical protocol, nonirradiated and irradiated *d*- and *l*-quartz blanks were separately dried and derivatized following the same procedure, as described for the above samples and analyzed (1 µL) by GC×GC-TOFMS. The volume of the blank aliquots was the same as for the corresponding infiltration experiment. No contamination was observed.

### 2.5. GC×GC-TOFMS Analysis–Instrumental Conditions

The enantioselective multidimensional analysis was carried out by a GC×GC Pegasus instrument coupled to a TOFMS (LECO, St. Joseph, MI, USA). The TOFMS system operated at a storage rate of 150 Hz, with a 50–400 amu mass range, a detector voltage of 1650 V, and a solvent delay of 15 min. Ion source and injector temperatures were set to 230 °C and the transfer line to 240 °C. The column set consisted of a Varian–Chrompack Chirasil-l-Val column (25 m × 0.25 mm, 0.12 µm film thickness, Agilent-Varian, Santa Clara, California, US) in the first dimension and a DB-Wax in the second dimension (1.3 m × 0.1 mm, 0.1 µm film thickness) connected by a siltite µ-union. 2-propanol and chloroform (sample solvents) were used as washing solvents for the injection needle. Helium was used as carrier gas with a constant flow of 1 mL min^−1^. All samples were injected in a 1:20 split mode.

For the analyses of samples containing alanine, the temperature of the primary oven was held at 40 °C for 1 min, increased to 80 °C at a rate of 10 °C min^−1^, and held for 5 min, followed by an increase to 120 °C at 2 °C min^−1^ and finally to 190 °C at 10 °C min^−1^ and held for 6 min. The temperature of the secondary oven was held at 65 °C for 1 min, increased to 105 °C at a rate of 10 ° C min^−1^ with a 5-min hold time, followed by an increase to 145 °C at 2 °C min^−1^ and, finally, to 190 °C at 8 °C min^−1^.

For the analyses of samples containing leucine, the temperature of the primary column was held at 40 °C for 1 min, then increased to 105 °C at a rate of 10 °C min^−1^ and held for 5 min, followed by an increase to 140 °C at 2 °C min^−1^ and finally to 190 °C at 10 °C min^−1^ and held for 5 min. The secondary oven used the same temperature program with a constant temperature offset of 20 °C.

For the mixed alanine–leucine samples, the temperature of the primary column was held at 40 °C for 1 min, increased to 80 °C at a rate of 10 °C min^−1^ and held for 5 min, then by an increase to 170 °C at 2 °C min^−1^, and finally to 190 °C at 10 °C min^−1^ and held for 5 min. The secondary oven used the same temperature program with a constant temperature offset of 25 °C. The modulator used a temperature offset of 40 °C compared to the primary oven temperature and a modulation period of 5 s was applied to all samples.

### 2.6. Enantiomeric Excess Determination 

The enantiomeric excess *ee* which expresses the excess of one enantiomer over the other is defined as *ee* = (E_1_ − E_2_)/(E_1_ + E_2_) where *E*_1_ and *E*_2_ are the quantities of the enantiomers. In practice, *ee* is often quoted as a percentage, defined as %*ee* = 100 × (E_1_ − E_2_)/(E_1_ + E_2_). Thus, a racemate is expressed as *ee* = 0%. While amino acid standards are commercially available as racemic mixtures, their expected equimolar ratios determined by chromatographic techniques are generally unequal to 0% (*ee* ≠ 0%) due to (*i*) intrinsic manufacture effects of the racemic standards itself; (*ii*) differential instrument responses for each enantiomer such as stereoselective signal suppression; or (*iii*) erroneous chromatographic quantitation caused by the co-elution of chemical impurities with only one enantiomer spuriously increasing peak areas, poor enantioresolution (*R*_S_ < 1.5), or peak tailing. To best account for any non-racemic composition measurements for analytes which are, in fact, racemic and vice versa, the non-zero *ee* of the racemic alanine and leucine standard, respectively, were determined as follows: %*ee*_ref_ = 100 × [*A*(l-*AA*_ref_) − *A*(d-*AA*_ref_)]/[*A*(l-*AA*_ref_) + *A*(d-*AA*_ref_)](1)
where *A* is the peak area of the GC×GC signal and *AA* is the corresponding amino acid. The %*ee* values of the alanine and leucine samples after the adsorption experiments on homochiral quartz sand were calculated analogously by subtracting the %*ee*_ref_ of the respective amino acid in the standard solutions used as references: %*ee_AA_* = 100 × [*A*(l-*AA*) − *A*(d-*AA*)]/[*A*(l-*AA*) + *A*(d-*AA*)] − %*ee*_ref_(2)

## 3. Results

A summary of the results of the GC×GC-TOFMS analyses of recovered interstitial liquids is given in Table 3 and Figure 3. Both amino acids, Ala and Leu, were enriched in the d-enantiomer (*ee*_L_ < 0), regardless of the handedness of the quartz sand. Moreover, the original hypothesis that capillary tubes on the quartz surface, produced by etching of uranium fission tracks, may enhance the overall enantioselective adsorption is partly disproved by our experiments.

First, we examined the absolute values of %*ee*_L_, regardless of their algebraic sign. We measured six %*ee*_L_ values for Ala, and additional six %*ee*_L_ values for Leu. The |%*ee*| for Ala in six independent experiments ranges from 1.1 to 15.2%. These %*ee* values are significantly above the 3*σ* standard deviations (*n* = 3). They demonstrate that substantial racemate resolution can be expected in interstitial pores, along grain boundaries or small fractures in a local quartz-bearing environment. The |%*ee*| for Leu in six independent experiments ranges from 3.5 to 55.6%. These values are even higher than those for Ala. They substantiate the foregoing conclusion that random quartz fractures may produce significant racemate resolution on a local scale. This statement holds true for polygranular quartz occurrences with a chiral enrichment of either *d*- or *l*-quartz.

Second, we examined the influence of thermal neutron irradiation, induced fission tracks, and radiation damage, in general, on the observed *ee* values. In the case of Ala, the mean %*ee*_L_ value in the three experiments using nonirradiated quartz is −4.0% (spread between −2.3% and −6.8%), and the mean %*ee*_L_ value with irradiated quartz is −9.7% (spread between −1.1 and −15.2%). These results indicate an overall stronger enantioselective adsorption of l-Ala on irradiated quartz compared with nonirradiated quartz. One exception is the experiment of the pure racemic alanine solution absorbed on *d*-quartz were the %*ee*_L_ of −2.92% for nonirradiated quartz decreased to −1.09% for irradiated quartz. Thus, the results for Ala are ambiguous and will be discussed later.

In the case of Leu, the mean %*ee*_L_ value of three independent experiments with nonirradiated quartz is −28.8% (spread between −6.8 and −55.6%), and the mean %*ee*_L_ value with irradiated quartz is −9.0% (spread between −3.5 and −12.6%). The l-enantiomer of Leu adsorbs distinctly stronger on nonirradiated quartz than on irradiated quartz, without any exception. In all experiments with either *d*- or *l*-quartz, the adsorption of the l-enantiomers on nonirradiated quartz was about two or even four times higher than on irradiated quartz. Strong radiation damage clearly reduces enantioselective adsorption of l-Leu onto the quartz surface or in any superficial structures. Thus, the hypothesis of Hejl is disproved for leucine [20]. This observed reduction in enantioselective adsorption on irradiated quartz might be due to metamictization of the crystal lattice. Radiation dose-dependent destruction of the crystalline structure, up to amorphous consistency, is well known for zircon [43]. Atomic scale observations by transmission electron microscopy (TEM) have shown that the core zone of latent fission tracks in zircon is amorphous [44]. If this amorphous domain is not completely removed by etching, the walls of the fission track capillary tube will keep a certain metamictization after etching. Although the atomic structure of latent fission tracks in quartz has not yet been investigated, a superficial metamictization by heavy ion bombardment from adjacent uraniferous minerals, such as zircon, cannot be excluded. This kind of radiation damage would explain the alteration of enantioselective adsorption properties.

Comparing the ratios of the peak areas of the amino acid and the internal standard (A_AA_/A_IS_) before and after the adsorption experiments on enantiomorphic quartz allowed us to evaluate the adsorption efficiencies of each experiment (Table 3). Starting from an initial amino acid concentration of 10^−3^ M (sum of both enantiomers), the recovered quantities were found to range from 1.6 × 10^−4^ M for the experiments using irradiated *l*-quartz up to 4.3 × 10^−4^ M for the nonirradiated *d*- and *l*-quartz samples. Thus, less than 15% to 25% of the amino acids were recovered from irradiated *l*- and *d*-quartz, respectively, whereas almost 50% of amino acids were desorbed from the nonirradiated enantiomorphic quartz samples when using individual amino acid solutions. While these results indicate an increased adsorption affinity of irradiated over nonirradiated enantiomorphic quartz due to irradiation-induced fission tracks—that was not found to directly correlate with enhanced enantioselective adsorption, no significant adsorption selectivity between alanine and leucine was observed. The results of the competitive adsorption experiments are less obvious with very similar adsorption affinities of the irradiated and nonirradiated *d*-quartz but inverse trends in enantioselectivity for alanine and leucine. While the %*ee*_L_ of alanine significantly decreased to −15% on irradiated *d*-quartz, i.e., enhanced adsorption of d-alanine on irradiated *d*-quartz, d-leucine adsorbed more efficiently on nonirradiated *d*-quartz compared to irradiated *d*-quartz. It is currently uncertain whether these differences in enantioselectivity of alanine and leucine in the combined mixture are due to systematic or pure random effects. We note, however, that, in preparation of the incubation experiments of the alanine–leucine mixtures, the quartz sand was filled using thin tubes, whereas small funnels were used to prepare the adsorption experiments of the individual amino acids. This may have eventually entailed a difference in the packing density of the grains and thus differences in the cumulative pore volume. The grain packing in the funnels corresponds much better to the textural conditions in natural sediments and will be chosen as a preferred experimental design for future adsorption experiments. Further competitive enantioselective adsorption experiments are required to study the profound impact of intimate binding between enantiomorphic quartz and amino acids of different size, charge, and polarizability, as well as mutual effects between different molecular species during competitive adsorption. 

## 4. Discussion

In our experimental investigation, the measured %*ee* values of both amino acids, Ala and Leu, were always enriched in the d-enantiomer (l-%*ee* < 0). Originally, we had assumed that sand grains have a nearly isometric form corresponding approximately to a sphere. With regard to the crystal lattice, all possible spatial orientations of crystalline planes occur with similar likelihood on the surface of the sphere. Even irregular planar orientations which cannot be denominated by Miller–Bravais indices would have the same frequency than any other plane. Indeed, sand grains of aeolian dunes, beach sands, or fluvial sediments are often well rounded [45]. Most quartz grains exhibit a spherical or ellipsoidal shape. If two monocrystalline quartz grains consist of quartz with opposite handedness (*d* and *l*, respectively), their overall chiral adsorption properties of their surfaces are expected to produce bulk-*ee* with opposite handedness on both grain surfaces. This assumption is still valid when a quasi-infinite number of homochiral grains with irregular forms is considered—provided that these irregular forms have no preferential orientation relative to the crystal lattice. Quartz has no preferential cleavage planes and mainly produces conchoidal fragments at rupture. This well-known fact seems to support the foregoing assumption.

However, the interstitial liquid of both enantiomorphic quartz crystals was always enriched in the d-enantiomer and no mirror effect was observed. Moreover, both l-Ala and l-Leu enantiomers tend to adsorb much stronger on *l*-quartz (natural) than on *d*-quartz (synthetic z-bar). We suppose that this phenomenon is due to different frequency distributions of the crystallographic orientation of fracture planes in the z-bar and the natural mountain quartz. The longitudinal axis of the z-bar is perpendicular to the crystallographic c-axis, while the longitudinal axis of the mountain crystals is parallel to the c-axis (Figure 4). These differences in the crystal habitus may have caused a dissimilar behavior at rupture. When a longitudinal homogenous material without preferential cleavage is broken with a hammer, one would expect that the predominant factures are mainly perpendicular to the longitudinal axis. These first fractures may have influenced the further ruptures in course of crumbling with the disk mill.

With regard to future adsorption experiments, we suggest and intend the use of *d*- and *l*-quartz with similar habitus for the production of homochiral sand—i.e., natural quartz crystals from the same location, with similar size and form. By this procedure, similar frequency distributions of crystallographic fractures orientations in both *d*- and *l*-quartz could be obtained. Random effects will be minimized, and the resulting bulk adsorption on the grain surfaces of *d*- or *l*-quartz will produce rather ‘symmetrical’ *ee* with opposite sign. 

Overall, leucine showed higher racemate resolution on quartz with a maximum %*ee* of 55.6% compared with alanine in our experiments. Molecular size-related effects may be responsible for the observed trend and future investigations on various individual amino acid solutions covering different molecular sizes or branching, and different functional groups (aliphatic, aromatic, acidic, basic, etc.) are required to study size- and polarization-related effects on the enantioselective adsorption behavior. Moreover, additional competitive adsorption experiments are required for investigating the interplay of amino acid mixtures and the enantioselective adsorption forces of enantiomorphic minerals to illuminate prebiotically plausible systems.

## 5. Conclusions

The enantioselective adsorption of alanine and leucine with %*ee* values ranging from 1% to 55% was observed to occur on homochiral quartz sand. The l-enantiomer of both investigated amino acids was preferably adsorbed regardless of the handedness of the enantiomorphic quartz. We currently explain the lack of mirror symmetry breaking in our experiments with the different crystal habitus of the synthetic z-bar of *d*-quartz and the natural mountain crystals of *l*-quartz. Future experiments using *d*- and *l*-quartz with similar habitus to minimize random effects are required to conclude on the possible mirror symmetry breaking stimulus through amino acid adsorption on enantiomorphic minerals with opposite handedness.

Despite the absence of %*ee* values of the opposite sign, there are consistent trends in our collected data, such as natural *l*-quartz, showed stronger enantioselective adsorption affinities than synthetic *d*-quartz for both amino acids. Moreover, strong radiation damage clearly reduced the enantioselective adsorption of leucine, while no clear conclusion can be drawn for the results on alanine regarding nonirradiated vs. irradiated quartz. Finally, the desorption capacity of alanine and leucine was equally strong in each of the corresponding experiments using individual amino acid solutions as well as using the combined amino acid mixture, i.e., no amino acid selectivity in terms of pure adsorption of alanine vs. leucine onto identical quartz surfaces. 

## Figures and Tables

**Figure 1 life-11-01222-f001:**
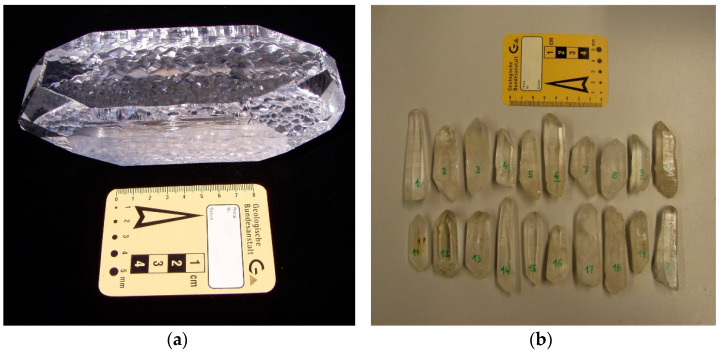
Enantiomorphic crystals of *d*- and *l*-quartz. (**a**) Synthetic z-bar of *d*-quartz produced by hydrothermal growth on a rectangular seed plate with a total mass of 1.473 kg. Its crystallographic c-axis is orientated perpendicular to the main elongation of the crystal and perpendicular to the table plane.; (**b**) Twenty natural quartz crystals from an open fissure in Madagascar. Ten of them were found to be single crystals without twinning nor intergrown with smaller crystals.

**Figure 2 life-11-01222-f002:**
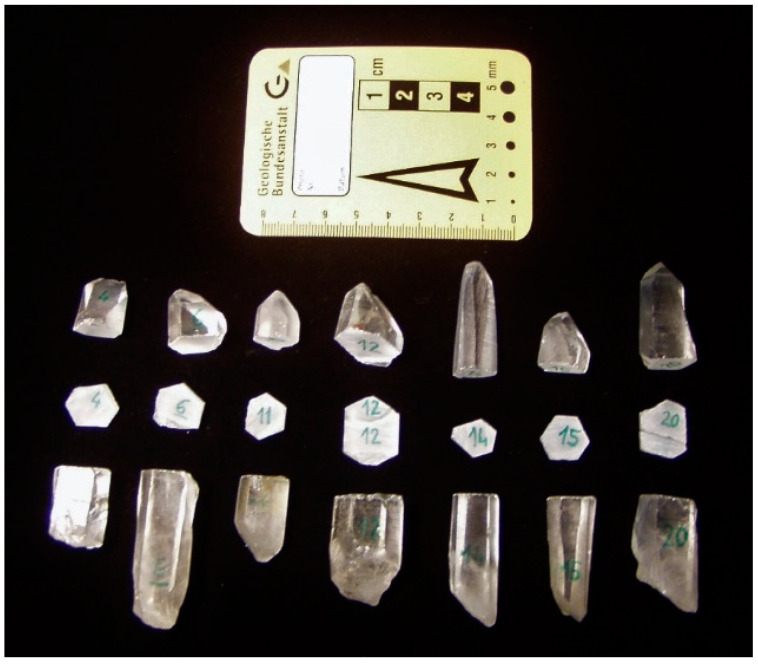
Some non-twinned mountain crystals with ca. 3-mm-thick sheets that were cut perpendicular to the crystallographic c-axis.

**Figure 3 life-11-01222-f003:**
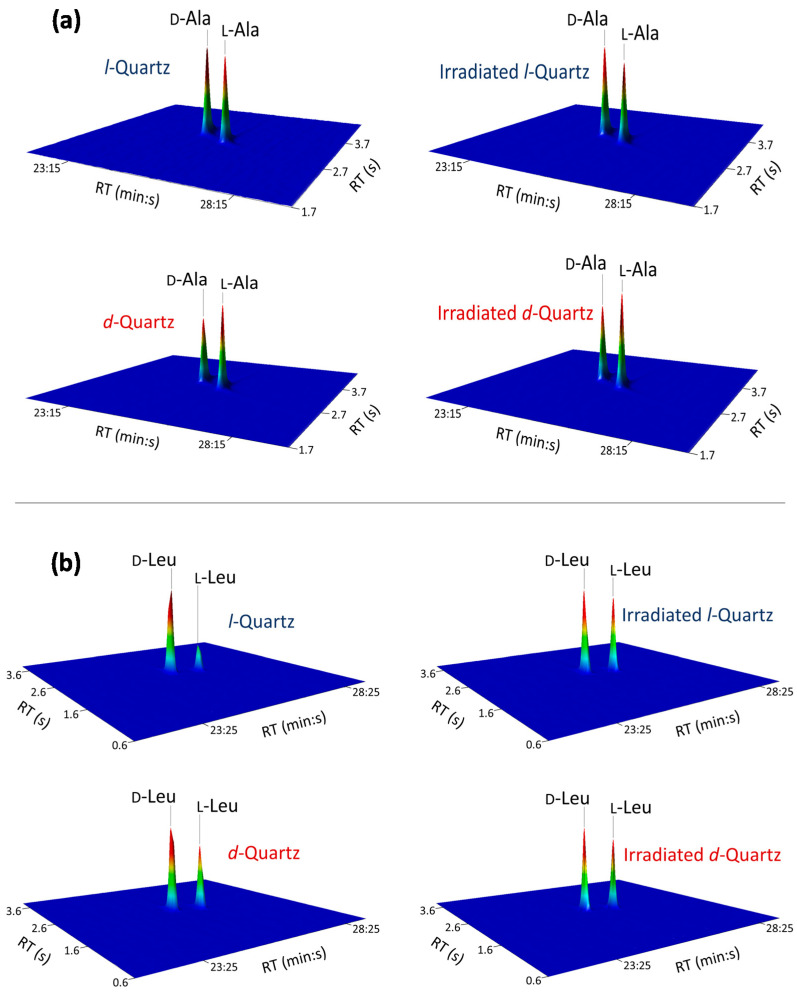
Enantiomorphic quartz induced enantiomeric excesses. Close-up view of the multidimensional enantioselective gas chromatographic analysis of (**a**) alanine and (**b**) leucine enantiomers after adsorption on nonirradiated and irradiated *l*-quartz (blue) and *d*-quartz (red).

**Figure 4 life-11-01222-f004:**
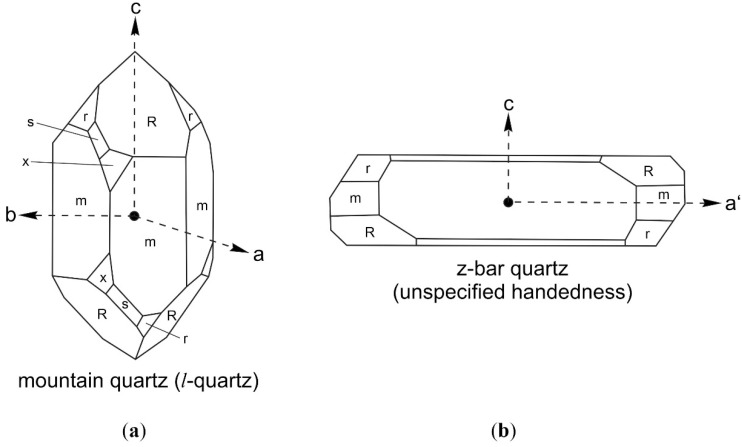
Orientations of crystallographic axes and principal crystal planes m, r, R, s and x after Cheng et al. [46]. (**a**) Schematic representation of l-quartz with space group P3121. The position of the faces of the trigonal trapezohedron x {6 $\bar{1\;}\;\bar{5\;}$ 1}and the trigonal bipyramid s {2 $\bar{1\;}\;\bar{1\;}$ 1} reflects the handedness of the crystal; (**b**) The longitudinal axis a’ of the z-bar quartz is orthogonal to a prism face m and forms an angle of 30° with the a-axis. The z-bar used for this investigation was *d*-quartz (Figure 1), but its handedness cannot be deduced from its morphology.

**Table 1 life-11-01222-t001:** Polarimetric measurements on quartz slices perpendicular to the crystallographic c-axes. Slices 4, 7, 11, 12, 14 and 20 were found to be *l*-quartz. The corresponding crystals were used to produce *l*-quartz sand.

Quartz Slices Normal to C Axis	Thickness [nm]	Optical Rotation [°]	${\left[\alpha \right]}_{D}^{20}$ [° mm^−1^]
Slice 4	2.99	−67.998	−22.74
Slice 6	3.35	+72.545	+21.66
Slice 7	3.02	−64.213	−21.26
Slice 9	3.16	+70.282	+22.24
Slice 11	2.91	−62.330	−21.42
Slice 12	3.07	−67.043	−21.84
Slice 14	3.16	−69.470	−21.98
Slice 15	3.22	+72.254	+22.44
Slice 16	3.17	*not measurable* ^1^	-
Slice 20	2.92	−63.292	−21.68
z-bar quartz	3.16	+68.693	+21.74

^1^ Slice 16 could not be measured because of internal fractures.

**Table 2 life-11-01222-t002:** Chemical composition of the zircon powder measured by X-ray fluorescence with major elements in oxide concentrations. Components are listed in descending order of concentrations.

Chemical Components	Concentration [wt %]	LLD ^1^ [ppm]
ZrO_2_	64.12	91.8
SiO_2_	33.44	403.1
Fe_2_O_3_	0.539	191.0
TiO_2_	0.372	61.5
Ce	0.284	51.6
P_2_O_5_	0.249	45.1
Al_2_O_3_	0.222	48.7
SO_3_	0.138	39.6
Cr	0.120	24.3
Na_2_O	0.084	37.4
CaO	0.062	45.9
U	0.049	22.9
MgO	0.049	50.0
MnO	0.028	14.2
Th	0.028	17.8
K_2_O	0.019	37.1
Sc	0.013	20.0
La	0.009	37.2
Pb	0.009	31.6
Ba	0.008	54.5
Cl	0.004	12.6
Ni	0.004	9.8
V	0.003	21.9
Sum	99.85	

^1^ Lower limit of detection.

**Table 3 life-11-01222-t003:** Enantiomeric excesses %*ee*_L_ of alanine and leucine after adsorption on nonirradiated and irradiated enantiomorphic *d*- and *l*-quartz. The recovered amino acid quantity is specified for each adsorption experiment as well as the recovered quantity normalized to the estimated total specific surface area of quartz (954 m^2^).

Amino Acid	%*ee*_L_-*d*-Quartz		%*ee*_L_-*l*-Quartz	
	Nonirradiated	Irradiated	Nonirradiated	Irradiated
alanine	−2.92 ± 0.26 ^2^	−1.09 ± 0.10 ^3^	−6.75 ± 0.32 ^2^	−12.82 ± 0.48 ^5^
alanine ^1^	−2.34 ± 0.45 ^3^	−15.21 ± 0.74 ^4^	-	-
leucine	−23.93 ± 0.35 ^2^	−10.86 ± 1.24 ^3^	−55.63 ± 0.46 ^2^	−12.58 ± 0.50 ^5^
leucine ^1^	−6.75 ± 0.52 ^3^	−3.57 ± 0.55 ^4^	-	-

^1^ Competitive adsorption experiment using a mixture of alanine and leucine. ^2^ c ≈ 4.3 × 10^−4^ M ≈ 4.5 × 10^−7^ M m^−2^. ^3^ c ≈ 3.2 × 10^−4^ M ≈ 3.4 × 10^−7^ M m^−2^. ^4^ c ≈ 4.0 × 10^−4^ M ≈ 4.2 × 10^−7^ M m^−2^. ^5^ c ≈ 1.6 × 10^−4^ M ≈ 1.7 × 10^−7^ M m^−2^.

## Data Availability

All data analyzed during this study are included in this published article. The datasets generated or analysed during the current study are available from the corresponding authors upon request.

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
