# Peer review of "Racemate Resolution of Alanine and Leucine on Homochiral Quartz, and Its Alteration by Strong Radiation Damage"

_life, 2021, doi:10.3390/life11111222_

Round 1
Reviewer 1 Report
Dear Editor, Dear Authors,
the manuscript
"Racemate resolution of alanine and leucine on homochiral quartz, and its alteration by strong radiation damage" (life-1413494)
represents a very interesting study and its results seem to be relevant in context of the origin of chirality. However, I miss consistent trends in the results and some details of interpretation. As I really like the concept of this work (research question, study design, type of analyses), I would be happy to see this Manuscript getting published but in this present form, I recommend to majorly revise it according to my comments below.
MAJOR ISSUES
~~~~~~~~~~~~
What is the take-home message about enantioseparation of these experiments? Is it stronger for irradiated or nonirr. quartz? Is it stronger for d- or for l-quartz? I don't see any coherent (but mixed) trends to get interpreted from these results.
Please discuss the influence of amino acid concentration / proportion of amino acid concentration to quartz amounts. Why have the authors chosen a concentration of 10–3 M (sum of both enantiomers)? Is this concentration relevant for an early Earth context?
How does radionuclic radiation affect the chemistry of amino acids? Do we observe radiation-induced racemization on the level of amino acid molecules or based on the quartz' surface modification (as suggest in Lines 391-401) here?
Why is there always an enrichment of the d amino acid?
DETAILED ISSUES
~~~~~~~~~~~~~~~
Line 18-19
"Our previous hypothesis on the enhanced enantioselectivity due to uranium-induced fission tracks could not be validated. " --> This isolated sentence is by no means helpful. Why did the original hypothesis fail??
Line 19-22
"In general, the natural l-quartz showed stronger adsorption affinities than synthetic d-quartz without any significant trend in amino acid selectivity. Surprisingly, the L-enantiomer of both amino acids alanine and leucine was preferably adsorbed regardless of the handedness of the enantiomorphic quartz sand" --> (i) Aren't these two sentence contradictory? (ii) Do the results not show a preference in d-amino acid adsorption (higher ee)?
Line 22/Line 22
Please remove subjective terms such as "surprisingly" or "unexpected"
Line 24-25
Future aims can be mentioned at the end of the Manuscript but should be removed from the Abstract. Furthermore, I miss a sentence in the Abstract about the radiation alteration results as indicated in the title.
Line 34-35
What is meant by "chemical evolution"? This term could be interpreted as an ongoing process. Please be more specific.
Line 42
Please define chirogenesis
Line 44
Again, please stay objectively ("worth to be considered")
Line 224
Stay consistently with capital or noncapital letters for D/d and L/l.
Line 347
Surprisingly (see above)
Line 362 / Table 3
* Why is there the opposite effect that alanine shows stronger enantiosepation for irradiated d-quartz when probing a mixed ala-leu solution, compared to single compound experiments and leucine in mixture, in which nonirradiated quartz shows stronger enantioseparation?
* Why is the effect different for ala for nonirr. vs irradiated d- and l-quartz? For leu, there is stronger enantioseparation for nonirr. d- and l-quartz while for ala, it is stronger for nonirr quartz only in d form but weaker for the nonirr. l form.
* Why is the effect in enantioseparation stronger for leucine than for alanine (difference in %ee for nonirr. vs irr. quartz for both d- and l-quartz)? Is there a molecular size-related effect?
Line 362-386 / Table 3
I am wondering if the numbers from Table 3 and the text afterwards do not match with each other. Please clarify that.
Table 3
d-quartz l-quartz
nonirr irr nonirr irr
ala -2.92 -1.09 -6.75 -12.82
ala (mix) -2.34 -15.21 - -
leu -23.93 -10.86 -55.63 -12.58
leu (mix) -6.75 -3.57 - -
Text
ala "mean [...] nonirradiated quartz is -4.0%"
but mean of [-2.92 , -6.75] is -4.835
"mean [...] irradiated quartz is -9.7%
but mean of [-1.09 , -12.82] is -6.955
leu "mean [...] nonirradiated quartz is -21.8%"
but mean of [-23.93 , -55.63] is -39.78
"mean [...] irradiated quartz is -8.9%
but mean of [-10.86 , -12.58] is -11.72
Line 370
Why shall? Please either write what you have done or remove it
Line 391-401
If this explanation would hold, we would observe the same effect for ala, right? Why is it then different for ala?
Line 424
Again, please remove any subjective opinions and stay on objective facts (here "unexpected")
Line 427
"with the same frequency" --> similarly likely
Like 442
"unexpected"
Line 440-441
"Moreover, both L-Ala 440 and L-Leu enantiomers tend to adsorb much stronger on l-quartz than on d-quartz" --> Isn't this contradictory to your results that show no difference between l- and d-quartz enantioselectivity?
Author Response
I am writing with reference to our Life manuscript life-1413494 entitled “Racemate resolution of alanine and leucine on homochiral quartz, and its alteration by strong radiation damage”. We highly appreciate all the time that you have dedicated to deliver a constructive review of our manuscript and we are grateful for the opportunity to submit a revised manuscript incorporating as best as possible, and where appropriate, your comments as well as the comments of the other two reviewers. We provide in the attached letter point-by-point responses to the reviewers’ reports. Changes in the manuscript have been highlighted in grey.
Reviewer 1
Dear Editor, Dear Authors,
the manuscript "Racemate resolution of alanine and leucine on homochiral quartz, and its alteration by strong radiation damage" (life-1413494) represents a very interesting study and its results seem to be relevant in context of the origin of chirality. However, I miss consistent trends in the results and some details of interpretation. As I really like the concept of this work (research question, study design, type of analyses), I would be happy to see this Manuscript getting published but in this present form, I recommend to majorly revise it according to my comments below.
MAJOR ISSUES
~~~~~~~~~~~~
*What is the take-home message about enantioseparation of these experiments? Is it stronger for irradiated or nonirr. quartz? Is it stronger for d- or for l-quartz? I don't see any coherent (but mixed) trends to get interpreted from these results.
Response: We do agree with the reviewer that our first results on the enantioselective adsorption of amino acids on homochiral quartz are somehow unexpected. We do not observe the anticipated mirror symmetry breaking using opposite handed quartz which we can currently only explain with the difference in crystal properties (using a synthetic z-bar of d-quartz and natural mountain crystals of l-quartz). However, we do see some consistent trends such as 1) stronger enantioselectivity of natural l-quartz compared with synthetic d-quartz for both amino acids, 2) no adsorption selectivity between alanine and leucine, and 3) reduced enantioselectivity after irradiation for leucine.
Based on the reviewer’s comment, we added a conclusion paragraph to summarize our major findings and hope this will provide an adequate take-home message.
*Please discuss the influence of amino acid concentration / proportion of amino acid concentration to quartz amounts. Why have the authors chosen a concentration of 10–3 M (sum of both enantiomers)? Is this concentration relevant for an early Earth context?
Response: The chosen concentration has no direct relevance for prebiotic chemistry/early Earth but has instead be chosen to accurately determine the recovered amino acid quantities and induced enantiomeric excesses. Considering an estimated total surface area of quartz of about 950 m2 in each of our experiment, the concentrations are sufficiently diluted to allow for explicit interaction (adsorption) of amino acids with the quartz surface.
We added the following sentence to the manuscript: “This amino acid concentration has been chosen to allow for accurate quantitation of recovered amino acids.”
*How does radionuclic radiation affect the chemistry of amino acids? Do we observe radiation-induced racemization on the level of amino acid molecules or based on the quartz' surface modification (as suggest in Lines 391-401) here?
Response: We have not investigated the effect of radiation on amino acids (radiation-induced racemization) but only the radiation effect on the possibility of enhancing the enantioselective chromatographic adsorption properties of quartz. Amino acids were not irradiated in our experiments. Yes, the referee is right, the reduced enantioselectivity (smaller absolute %ee values) of irradiated quartz is currently explained by the modifications of the crystalline structure (Lines 391-401).
*Why is there always an enrichment of the d-amino acid?
Response: That is a very good question that we have asked ourselves as well, as we expected to obtain mirror symmetry results in terms of %ee for opposite handed minerals. However, we currently propose that the different behavior at rupture of natural l-quartz and synthetic z-bar d-quartz may have influenced the grain habitus of the quartz sand used in our adsorption experiments (please see discussions). To avoid this problem in the future, we propose to use homochiral sand from similar (natural) sources of opposite handedness. Unfortunately, we did not find any supplier that can provide synthetic d- and l-quartz.
DETAILED ISSUES
~~~~~~~~~~~~~~~
Line 18-19
"Our previous hypothesis on the enhanced enantioselectivity due to uranium-induced fission tracks could not be validated. " --> This isolated sentence is by no means helpful. Why did the original hypothesis fail??
Response: Thank you for your comment. We have added an additional explanation to the abstract: “Such capillary tubes in the near-surface structure of quartz have been proposed to increase the overall chromatographic separation of enantiomers, but no systematic positive correlation of accumulated radiation damage and enantioselective adsorption was observed in this study”
Line 19-22 "In general, the natural l-quartz showed stronger adsorption affinities than synthetic d-quartz without any significant trend in amino acid selectivity. Surprisingly, the L-enantiomer of both amino acids alanine and leucine was preferably adsorbed regardless of the handedness of the enantiomorphic quartz sand" --> (i) Aren't these two sentence contradictory? (ii) Do the results not show a preference in d-amino acid adsorption (higher ee)?
Response: Thank you very much for these questions. The referee is right, that our results show an overall preference for the adsorption of the L-enantiomer of alanine and leucine on both opposite enantiomorphic quartz samples (which was unexpected for us): “Moreover, the L-enantiomer of both investigated amino acids alanine and leucine was preferably adsorbed regardless of the handedness of the enantiomorphic quartz sand.”
Comparing the absolute induced enantiomeric excesses %ee after adsorption on l- and d-quartz – that are always in favor of the d-enantiomer – stronger enantioselectivity was observed for the l-quartz. The determined %ee reached values of up to -55% for leucine recovered from nonirradiated l-quartz compared with -24% in the experiment using nonirradiated d-quartz. Alanine showed the exact same trend. Stronger enantioselective adsorption affinities therefore relate to the absolute %ee values that were always found to be higher in the experiments with l-quartz compared with the corresponding experiments using d-quartz. The observed handedness (sign of ee) was always the same. We believe that our abstract provides a short summary on the overall results which are explained in much more detail in the results/discussion part of our manuscript. We have slightly rephrased the abstract to avoid any misinterpretation.
Line 22
Please remove subjective terms such as "surprisingly" or "unexpected"
Response: ok, we removed ‘Surprisingly’ and ‘unexpected’
Line 24-25
Future aims can be mentioned at the end of the Manuscript but should be removed from the Abstract. Furthermore, I miss a sentence in the Abstract about the radiation alteration results as indicated in the title.
Response: Thank you very much for your advice. We have removed the future aims from the abstract and added further information on the radiation alteration: “…but no systematic positive correlation of accumulated radiation damage and enantioselective adsorption was observed in this study.”
Line 34-35
What is meant by "chemical evolution"? This term could be interpreted as an ongoing process. Please be more specific.
Response: Chemical evolution is among one of the theories of life’s origin and favored by most scientists working in this field today. The central premise of chemical evolution specifies that life arose naturally from abiotic matter (nonliving). Chemical evolution doesn’t occur suddenly; instead, it proceeds more gradually, eventually building complex structures from simpler ones. This modern theory then suggests that life originated on Earth by means of a rather slow evolution of nonliving matter. How slowly and when precisely we are unsure. We have added a short explanation and two more refs: “chemical evolution. While the historical details are still unclear, it is well accepted that chemical evolution did not occur suddenly; instead a gradual transition from simple organic building blocks including chiral molecules toward polymeric and supramolecular assemblies has built the fundaments for self-assembling, self-sustaining interactive systems with emerging patterns that have ultimately evolved into what could be considered as living entities [Eschenmoser 2007; Krishnamurthy & Hud, Chem Rev 2020].”
Line 42
Please define chirogenesis
Response: The study of chirogenesis of organic molecules elucidates the origin of the homochirality phenomenon of life’s biomolecules on Earth. We have added a short explanation to the introduction: “Several mechanisms capable to entail molecular symmetry breaking leading to biomolecular homochirality on Earth, so-called chirogenesis, have been proposed and investigated.”
Line 44
Again, please stay objectively ("worth to be considered")
Response: ok, we have rephrased the sentence to “…, has not yet been evaluated experimentally.”
Line 224
Stay consistently with capital or noncapital letters for D/d and L/l.
Response: We appreciate the referee’s comment, however, we would prefer to use the descriptors as they are in the manuscript. We have actively chosen the use of small capitals (small caps) for the stereochemical configuration of amino acids (d, l) whereas noncapital letters of l and d (in italics) are used for the handedness of minerals to respect the traditional nomenclature in stereochemistry and mineralogy. A unification of the typography would violate one of these existing nomenclatures.
Line 347
Surprisingly (see above)
Response: ok, we removed ‘Surprisingly’
Line 362 / Table 3
* Why is there the opposite effect that alanine shows stronger enantiosepation for irradiated d-quartz when probing a mixed ala-leu solution, compared to single compound experiments and leucine in mixture, in which nonirradiated quartz shows stronger enantioseparation?
Response: Thank you very much for this question. At the moment, we do not know whether these observed differences are systematic or due to random effects by some variations in handling the funnels or tubes. The sand with the mixed Ala-Leu solution was filled in thin tubes, while the pure compounds were treated in small funnels. This can eventually entail a difference in the packing density of the grains (cumulative pore volume). We have added a brief comment related to the competitive experiments to the main text.
“It is currently uncertain whether these differences in enantioselectivity of alanine and leucine in the combined mixture are due to systematic or pure random effects. We note, however, that in preparation of the incubation experiments of the alanine-leucine-mixtures, the quartz sand has been filled using thin tubes whereas small funnels have been used in preparing the adsorption experiments of the individual amino acids. This may have eventually entailed a difference in the packing density of the grains and thus differences in the cumulative pore volume.”
* Why is the effect different for ala for nonirr. vs irradiated d- and l-quartz? For leu, there is stronger enantioseparation for nonirr. d- and l-quartz while for ala, it is stronger for nonirr quartz only in d form but weaker for the nonirr. l form.
Response: Regarding the experiments with leucine, the radiation damage clearly reduces the enantioselective adsorption capacity of quartz (please see manuscript page 11). We explained this in the text with potential metamictization of the crystal lattice due to radiation-induced destruction of the crystalline structure. The results for leucine are consistent in this regard. Alanine, however, shows inconsistent data in terms of the evolution of ee in the experiments with and without irradiation. At this point we can only speculate, and additional experiments with similar d- and l- quartz samples as discussed in the manuscript are required to conclude on the results of alanine as well as the experiments using amino acid mixtures to understand competitive adsorption effects.
* Why is the effect in enantioseparation stronger for leucine than for alanine (difference in %ee for nonirr. vs irr. quartz for both d- and l-quartz)? Is there a molecular size-related effect?
Response: We can imagine that size- and polarization-related effects can play a role but based on these very first results such assumptions are too speculative for a well-sounded conclusion. This will be a subject for future investigations that should include amino acids of different size and functional groups (aliphatic, aromatic, acidic, basic, cyclic, etc.) in individual and combined solutions.
Line 362-386 / Table 3
I am wondering if the numbers from Table 3 and the text afterwards do not match with each other. Please clarify that.
Text:
Ala "mean [...] nonirradiated quartz is -4.0%" Response: This is correct.
but mean of [-2.92 , -6.75] is -4.835 Response: [-2.92, -2.34, -6.75] = -4%
"mean [...] irradiated quartz is -9.7% Response: This is correct.
but mean of [-1.09 , -12.82] is -6.955 Response: [-1.09, -15.21, -12.82] =-9.7%
leu "mean [...] nonirradiated quartz is -21.8%" Response: The corrected value is: -28.8%
but mean of [-23.93 , -55.63] is -39.78 Response: [-23.93, -6.75, -55.63] = -28.8%
"mean [...] irradiated quartz is -8.9% Response: This corrected value is: 9.0%
but mean of [-10.86 , -12.58] is -11.72 Response: [-23.93, -6.75, -55.63] = 9%
Response: Thank you very much for carefully checking the table and the corresponding text. We have carefully recalculated the ee values reported in the text, and we found that those for alanine were consistent with Table 3, but those for leucine were indeed partly wrong in the original version and have been corrected. The reported ‘spread’ in the text indicates the two extreme values of three measurements. The mean %eeL was always calculated with three values (i.e. arithmetic mean).
To avoid any confusion, we have slightly modified the text for alanine as follows:
“In the case of Ala, the mean %eeL value in the three experiments using nonirradiated quartz is -4.0%...” i.e. Ala mean [-2.92, -2.34, -6.75] nonirradiated quartz is -4.0%"
As well as for leucine: “In the case of Leu, the mean %eeL value of three independent experiments…”
Line 370 Why shall? Please either write what you have done or remove it
Response: ok, we removed ‘shall’
Line 391-401 If this explanation would hold, we would observe the same effect for ala, right? Why is it then different for ala?
Response: The results for Ala are indeed ambiguous and not fully understood. Size-related effects may eventually play a role, but without additional investigations on more species of amino acids this is still an open question.
Line 424 Again, please remove any subjective opinions and stay on objective facts (here "unexpected")
Response: ok, we have rephrased the sentence to: “In our experimental investigation, the measured %ee values of both amino acids, Ala and Leu, were always enriched in the d‑enantiomer (l-%ee < 0).”
Line 427 "with the same frequency" --> similarly likely
Response: ok, we have changed the original wording “…crystalline planes occur with the same frequency on the surface….” to “… crystalline planes occur with similar likelihood on the surface …”
Line 442 "unexpected"
Response: ok, we have removed ‘unexpected’.
Line 440-441 "Moreover, both L-Ala 440 and L-Leu enantiomers tend to adsorb much stronger on l-quartz than on d-quartz"--> Isn't this contradictory to your results that show no difference between l- and d-quartz enantioselectivity?
Response: The term stronger refers to the difference of absolute %ee of each amino acid after adsorption on l- and d-quartz, respectively (Table 3). For example, the measured %eeL of alanine on d-quartz was found to be -3% (nonirradiated) and -1% (irradiated), i.e. slightly more of the l-alanine enantiomer is adsorbed on d-quartz. While both l-quartz samples were still found to be enriched in the l-alanine enantiomer (so no difference in terms of enantioselectivity in between the two enantiomorphic quartz samples), the %eeL found for alanine after adsorption on l-quartz reached values of -7% (nonirradiated) and -13% (irradiated), i.e. much stronger adsorption of l-alanine on l-quartz compared with d-quartz leading to an higher excess of d-alanine in the remaining solution. The same trend was observed for leucine adsorbed on d- and l-quartz.

Reviewer 2 Report
The authors conducted adsorption experiments using racemic amino acids to d- and l-quartz to evaluate enantio selective adsorption. The results are unexpected, as authors mentioned, and I am not very convinced the results and the interpretations. However, the result is the result and I understand it is not always easy to explain such phenomenon, and thus I am not going to complain about it. Please also find my comments below.
Line 18, “Our previous hypothesis on the enhanced enantioselectivity due to uranium-induced fission tracks could not be validated.”
This sentence comes out of the blue. Please add more explanations.
Line 75, “Detrital zircons in gneisses of ...” This also comes out of the blue. Please explain why you use zircons.
Line 137, “Afterwards this purified sand was heated for 3 hours at
500 °C in a muffle furnace.”
I guess this treatment was done for eliminating organic contaminations, but please clarify.
(same for line 186)
Line 194, “Thermal neutron irradiation of the filled tubes was performed...”
I don’t understand why you use zircon in addition to neutron irradiation. I thought zircon was added to get neutron irradiation to quartz. Then you conducted neutron irradiation from reactor. I may misunderstand, but why you need both zircon and neutron reactor?
Line 226, “then glowed for 3 h in a muffle furnace.”
Please add temperature.
Line 244, Please reconsider if the word “Interstitial” is appropriate here.
Lines 223-252, In these experiments, which quartz, natural or artificial, was used?
Line 257, “> 3 ppb total organic carbon”
Can you describe as “less than X ppb total...” rather than “over 3 ppb...” ?
Section 2.5, what is the purpose of second GC? Short explanation would be helpful.
Line 355, 3δ(delta) -> 3σ (sigma) ?
Table 3, Can you include recovery ratio or recovered quantity in Table 3? (not as footnote, since it is not easy to follow)
Line 371, tacks -> tracks
L441, “We suppose that this unexpected phenomenon is due to different frequency distributions of the crystallographic orientation of fracture planes in the z-bar and the natural mountain quartz.”
It is first time that you mentioned difference between natural and artificial quartz. Which results are from natural quartz (or artificial one)? Please clarify in the experimental and results sections.
Why don’t you add conclusion section?
Author Response
I am writing with reference to our Life manuscript life-1413494 entitled “Racemate resolution of alanine and leucine on homochiral quartz, and its alteration by strong radiation damage”. We highly appreciate all the time that you have dedicated to deliver a constructive review of our manuscript and we are grateful for the opportunity to submit a revised manuscript incorporating as best as possible, and where appropriate, your comments as well as the comments of the other two reviewers. We provide in the attached letter point-by-point responses to the reviewers’ reports. Changes in the manuscript have been highlighted in grey.
Reviewer 2
The authors conducted adsorption experiments using racemic amino acids to d- and l-quartz to evaluate enantioselective adsorption. The results are unexpected, as authors mentioned, and I am not very convinced the results and the interpretations. However, the result is the result and I understand it is not always easy to explain such phenomenon, and thus I am not going to complain about it. Please also find my comments below.
Line 18, “Our previous hypothesis on the enhanced enantioselectivity due to uranium-induced fission tracks could not be validated.” This sentence comes out of the blue. Please add more explanations.
Response: Thank you for your comment. We have added an additional explanation to the abstract: “Such capillary tubes in the near-surface structure of quartz have been proposed to increase the overall chromatographic separation of enantiomers, but no systematic positive correlation of accumulated radiation damage and enantioselective adsorption was observed in this study”
Line 75, “Detrital zircons in gneisses of ...” This also comes out of the blue. Please explain why you use zircons.
Response: These detrital zircons from the Yilgarn craton have not been used in our laboratory experiments but are a “mineralogical trace fossil” for the existence of granite and related plutonic rocks with high content of quartz. Such zircons as those described by WILDE et al. (2001) do not occur in basaltic crust (not on the Moon for example). They prove that quartz was abundant on the early Earth. An additional explanatory line has been added to the manuscript. Please see: “The presence of a granitic crust and thus the widespread occurrence of quartz on the early Earth are testified by detrital zircons in gneisses of the Yilgarn craton in Western Australia.”
Line 137, “Afterwards this purified sand was heated for 3 hours at 500 °C in a muffle furnace.”
I guess this treatment was done for eliminating organic contaminations, but please clarify.
(same for line 186)
Response: Indeed, this procedure has been carried out to eliminate biological contamination that can result in biases versus the l-enantiomer. We have added the following information to the experimental section: “…to minimize biological contaminations.” (line 141) and ”…to reduce the risk of biological contaminants such as natural l-amino acids.” (line 195)
Line 194, “Thermal neutron irradiation of the filled tubes was performed...”
I don’t understand why you use zircon in addition to neutron irradiation. I thought zircon was added to get neutron irradiation to quartz. Then you conducted neutron irradiation from reactor. I may misunderstand, but why you need both zircon and neutron reactor?
Response: Thank you for your comment. Indeed, this is a misunderstanding. Neither zircon alone (without neutron irradiation) nor neutron irradiation of pure quartz (without zircon) would produce the desired quantity of fission tracks. The natural (spontaneous) fission of 238U is a very rare event. It is 1.8 million times less frequent than the spontaneous alpha decay. With natural zircon we would have to wait several million years to produce a significant number of fission tracks on the quartz surface. On a side note, natural Uranium does not emit many free neutrons (almost nothing). Zircon is not a natural neutron source but only emits alpha particles and related gamma radiation by its natural radioactivity. Neutron irradiation in the reactor causes the fission of 235U (less frequent than 238U), and thus produces the desired fission tracks in quartz grains adjacent to uraniferous zircon. In our manuscript, we cite relevant literature in the introduction that provides this specific information on fission track dating such as WAGNER & VAN DEN HAUTE (1992) and by HEJL (2017).
However, to account for the referee’s comment we have added the following sentences to the experimental section: Note that the addition of zircon powder alone would not lead to enough fission tracks within an adequate time scale as the spontaneous fission of 238U is a very rare event. It would take several million years to produce a significant quantity of fission tracks on the surface of quartz. We thus accelerated the fission of 235U by subsequent neutron irradiation experiments of the mixed quartz/zircon powder.
Line 226, “then glowed for 3 h in a muffle furnace.”
Please add temperature.
Response: OK, we have added the temperature (500 °C) to the manuscript.
Line 244, Please reconsider if the word “Interstitial” is appropriate here.
Response: Thank you for suggesting changing ‘interstitial’. We have changed the wording as follows: “In this way, the aqueous solution containing the diluted amino acid depleted in one adsorbed enantiomer was rinsed out from the pores between the sand grains and recovered in a small beaker and then filled in vials with silicon screw caps.”
Lines 223-252, In these experiments, which quartz, natural or artificial, was used?
Response: We had to use both kinds of quartz as we could not commercially obtain synthetic l-quartz (see line 145). We have added this information also to the discussion section (line 460).
Line 257, “> 3 ppb total organic carbon”
Can you describe as “less than X ppb total...” rather than “over 3 ppb...” ?
Response: Thanks, absolutely right. We changed the relational operator from > to <.
Section 2.5, what is the purpose of second GC? Short explanation would be helpful.
Response: For the determination of enantiomeric excesses in the recovered amino acid solutions after adsorption, we used two-dimensional gas chromatography coupled to a TOFMS detector. A technique well implemented in our research consortium to accurately determine small ee values in complex samples. We have added two more refs to the manuscript including one review providing an overview on this analytical technique as well as the application of GC×GC to the analysis of enantiomer-enriched amino acids extracted from a meteorite.
Briefly, the limited sensitivity and resolving power of classical one-dimensional GC often leads to coelution and thus biases in the determination of small ee values in complex organic matrices. Using more advanced GC×GC-TOFMS with the most obvious advantage of providing much greater capacity for resolving constituents of complex samples, enantio-GC×GC analysis allowed to accurately quantify each amino acid enantiomer in our samples regardless of the background noise through physical resolution of enantiomers and the mineral matrix.
Line 355, 3δ(delta) -> 3σ (sigma)?
Response: Yes, the referee is right. Thanks. We changed 3δ to 3σ in the manuscript.
Table 3, Can you include recovery ratio or recovered quantity in Table 3? (not as footnote, since it is not easy to follow)
Response: Unfortunately, we cannot add the recovery rates directly to the table due to space constraints. We believe that the recovery rates as well as normalized recovery rates to the total specific surface area (SSA) do not require an additional table and that the trend of decreased amino acid quantity after irradiation of individual amino acid solutions is sufficiently clear, especially for the l-quartz experiments. Recoveries of alanine and leucine were identical in similar experiments with no evident adsorption selectivity for one of the two amino acids.
Line 371, tacks -> tracks
Response: OK, tacks has been replaced by tracks
L441, “We suppose that this unexpected phenomenon is due to different frequency distributions of the crystallographic orientation of fracture planes in the z-bar and the natural mountain quartz.”
It is first time that you mentioned difference between natural and artificial quartz. Which results are from natural quartz (or artificial one)? Please clarify in the experimental and results sections.
Response: Thank you very much for your comment. In addition to Table 3, we now also provide this information in the experimental part (line 232).
Why don’t you add conclusion section?
Response: Thank you very much for this remark that motivated us to add the following conclusions:
Enantioselective adsorption of alanine and leucine with %ee values ranging from 1% to 55% was observed to occur on homochiral quartz sand. The l-enantiomer of both investigated amino acids was preferably adsorbed regardless of the handedness of the enantiomorphic quartz. We currently explain the lack of mirror symmetry breaking in our experiments with the different crystal habitus of the synthetic z-bar of d-quartz and the natural mountain crystals of l-quartz. Future experiments using d- and l-quartz with similar habitus to minimize random effects are required to conclude on the possible mirror symmetry breaking through amino acid adsorption on opposite enantiomorphic minerals.
Despite the absence of %ee values of opposite sign, there are consistent trends in our collected data such as natural l-quartz showed stronger enantioselective adsorption affinities than synthetic d-quartz for both amino acids. Moreover, strong radiation damage clearly reduced the enantioselective adsorption of leucine, while no clear conclusion can be drawn for the results on alanine regarding nonirradiated vs irradiated quartz. Finally, the desorption capacity of alanine and leucine was equally strong in each of the corresponding experiments using individual amino acid solutions as well as using the combined amino acid mixture, i.e. no amino acid selectivity in terms of pure adsorption of alanine vs leucine onto identical quartz surfaces.

Reviewer 3 Report
The manuscript with title “Racemate resolution of alanine and leucine on homochiral quartz, and its alteration by strong radiation damage” written well in form of research article. The studies reported here deal with determination of adsorption capacity from liquid phase of enantiomeric forms of alanine and leucine by irradiated and not irradiated homochiral quartz particles. Author demonstrated that natural l-quartz possess stronger adsorption capacity than synthetic quartz. L-enantiomer of both amino acids was preferentially adsorbed regardless the nature of enantiomorphic quartz send. Though this manuscript is interesting in the field of selectivity of adsorbed amino acid on mineral surface based on their enantiomeric structure there are major revisions are needed prior to have this manuscript ready for submission.
The comments are reported below as follows:
- Authors reported results of XRF analysis of studied here materials but never made logical connection between XRF studies and key findings (selectivity on adsorption capacity of Leu and Ala);
- Line 280. Please specify what enantiomeric form of Leu and Ala you used as standard;
- In the table 3 authors reported that nonirradiated l-quartz retain 4.3E-4M of Leu while irradiated retains 1.6E-4M. In the discussion authors refer to orientation of crystallographic planes exposed to the surface (figure 4) but never come with clear characterization of planes. To operate with data dealing with crystallographic studies (in particular studies of planes exposed for adsorption) authors have to use appropriate methods of investigations. In my humble opinion lines 439-467 of discussion should be rewritten using strong and constructive arguments for data interpretation.
- In the experimental section authors mentioned that crystals were crushed to obtain crystal sends that later was used for adsorption and enantiomer transformation studies. Authors never took into consideration specific surface area (SSA) of final powder used for adsorption Ala and Leu. As a result of milling crystals SSA could be different and therefore also adsorption capacity and enantiomer transformation changes. Authors should measure and report SSA of milled crystals prior to interpret findings.
- Lines 405-409 determination of percentage of desorbed amino acids should be normalized to specific surface area. Normally BET method (by adsorption of N2) is used to determine SSA of powdered materials.
- Line 429-430 please add reference.
- Line 442-443 please use appropriate terms. It is not clear statement with following sentence “ frequency distribution of crystallographic orientation”. You might intended to say abundancy of crystallographic planes? Please specify.
- Figure 4 should be presented with identical axis orientation. Otherwise, it is hard to see the difference between those two crystals. Though in my humble opinion it is not worth to refer to ideal theoretical crystals because after milling their shape and area of planes exposed do not correspond to schemas presented in figure 4.
- Section of conclusion is absent. Please write conclusion of main findings.
Author Response
I am writing with reference to our Life manuscript life-1413494 entitled “Racemate resolution of alanine and leucine on homochiral quartz, and its alteration by strong radiation damage”. We highly appreciate all the time that you have dedicated to deliver a constructive review of our manuscript and we are grateful for the opportunity to submit a revised manuscript incorporating as best as possible, and where appropriate, your comments as well as the comments of the other two reviewers. We provide in the attached letter point-by-point responses to the reviewers’ reports. Changes in the manuscript have been highlighted in grey.
Reviewer 3
The manuscript with title “Racemate resolution of alanine and leucine on homochiral quartz, and its alteration by strong radiation damage” written well in form of research article. The studies reported here deal with determination of adsorption capacity from liquid phase of enantiomeric forms of alanine and leucine by irradiated and not irradiated homochiral quartz particles. Author demonstrated that natural l-quartz possess stronger adsorption capacity than synthetic quartz. L-enantiomer of both amino acids was preferentially adsorbed regardless the nature of enantiomorphic quartz send. Though this manuscript is interesting in the field of selectivity of adsorbed amino acid on mineral surface based on their enantiomeric structure there are major revisions are needed prior to have this manuscript ready for submission.
The comments are reported below as follows:
1) Authors reported results of XRF analysis of studied here materials but never made logical connection between XRF studies and key findings (selectivity on adsorption capacity of Leu and Ala);
Response: We agree with this comment. Indeed, the details of the zircon bulk chemistry are not relevant for any selective adsorption properties of neutron-irradiated quartz. We added Table 2 only for a better characterization of the natural material that we have used in our experiment. The reason why we have not used pure UO2 (uraninite) instead of zircon is that we needed a uniform distribution of uranium over the whole pore volume and the whole quartz surface. This would have been almost impossible with an extreme low quantity of pure UO2. Therefore, we needed a solid material with strongly ‘diluted’ uranium. In fact, only the U concentration and to some extent the Th concentration are relevant for the calculation of an appropriate neutron fluence and corresponding track density. When Th/U > 1 (not the case for our zircon powder) some fission-tracks produced by Th-fission (induced by fast or epithermal neutrons) can be expected too. But this detail has no importance in the context of this article. It is not worth to be discussed here (cf. textbooks).
Thus, Table 2 can be removed without losing relevant information. We leave this open to the editor’s decision.
2) Line 280. Please specify what enantiomeric form of Leu and Ala you used as standard;
Response: Thank you. We used racemic standards identical to the ones used for the adsorption experiments. We added ‘racemic’ to the manuscript.
3) In the table 3 authors reported that nonirradiated l-quartz retain 4.3E-4M of Leu while irradiated retains 1.6E-4M. In the discussion authors refer to orientation of crystallographic planes exposed to the surface (figure 4) but never come with clear characterization of planes. To operate with data dealing with crystallographic studies (in particular studies of planes exposed for adsorption) authors have to use appropriate methods of investigations. In my humble opinion lines 439-467 of discussion should be rewritten using strong and constructive arguments for data interpretation.
Response: Thank you for your comment that we believe is a misunderstanding. Figure 4 does not deal with planes specific for adsorption but shall only explain the different habitus (longitudinal axis) and fracture behavior of both kinds of crystals. The fracture surfaces of quartz are conchoidal (irregular) and in almost every case not crystallographically well defined planes (no Miller indices, i.e. no rational number in axis proportions). The reported recovery rates are not related to the displayed crystal faces of Figure 4. Quartz has no cleavability, and its random fractures are very similar to those of glass, i.e. not planar at all. The original form or habitus of the crystals may entail a certain bias – either subparallel or subperpendicular to the c-axis. At that point, we can only try to explain our results by considering the difference in crystal habitus of natural vs synthetic quartz. We hope that our future experiments sampling natural quartz with opposite handedness provide deeper insights into enantioselective adsorption and probably ee values of opposite sign.
4) In the experimental section authors mentioned that crystals were crushed to obtain crystal sends that later was used for adsorption and enantiomer transformation studies. Authors never took into consideration specific surface area (SSA) of final powder used for adsorption Ala and Leu. As a result of milling crystals SSA could be different and therefore also adsorption capacity and enantiomer transformation changes. Authors should measure and report SSA of milled crystals prior to interpret findings.
Response: Thank you very much for this comment. In fact, the quartz was not powdered but only granulated and sieved. The zircon was indeed milled to a very fine-grained powder but was fully removed by decantation after etching and repeated washing with bidistilled water. This zircon powder never encountered the amino acid solutions. We therefore consider the SSA of the zircon powder not to be relevant for the adsorption experiments. On the other hand, the granulated quartz used for the adsorption experiments is rather coarse-grained and almost equigranular (well sorted, grain-size between 0.25 and 0.5 mm). In this case, SSA can indeed be estimated from the grain-size or grain-size distribution (when not well sorted) with sufficient precision, as it is often applied in soil science and concrete investigations (cf. ERSAHIN et al., 2006). The roughness of grain surfaces does not change the order of magnitude.
We have added the following information and an additional reference to the manuscript: The specific surface area (SSA) of both kinds of quartz sand (l and d) can be deduced from their quasi equigranular grain-size (0.25 – 0.5 mm), using published data of granulometric investigations (Ersahin et al., 2006). Recovered amino acid quantities normalized to SSA (Table 3) were calculated with a SSA of 60 m2 g-1 and with a porosity of 40% that is typical for equigranular sand with angular grains.
5) Lines 405-409 determination of percentage of desorbed amino acids should be normalized to specific surface area. Normally BET method (by adsorption of N2) is used to determine SSA of powdered materials.
Response: Thank you very much for this suggestion. An estimation of the quantities normalized to the specific surface area has been added (Table 3). The BET method is indeed useful for very fine-grained materials with heterogeneous grain-size distribution but not necessary for quasi equigranular materials. Considering a specific surface area (SSA) from ref. Ersahin et al. (2006) of 60 m2 g-1 and a porosity of quartz sand of 40%, we estimate the total surface area in our experiment to be approximately 954 m2.
6) Line 429-430 please add reference.
Response: Thank you very much for suggesting to add a reference here. The newly added ref is: REINECK, H.-E.; SINGH, I.B. Depositional Sedimentary Environments; Springer Berlin-Heidelberg-New York, 1980; ISBN 3-540-10189-6.
7) Line 442-443 please use appropriate terms. It is not clear statement with following sentence “ frequency distribution of crystallographic orientation”. You might intended to say abundancy of crystallographic planes? Please specify.
Response: According to the reviewer’s suggestion, we modified our statement as follows: “… all possible spatial orientations of crystalline planes occur with similar likelihood…”
8) Figure 4 should be presented with identical axis orientation. Otherwise, it is hard to see the difference between those two crystals. Though in my humble opinion it is not worth to refer to ideal theoretical crystals because after milling their shape and area of planes exposed do not correspond to schemas presented in figure 4.
Response: The referee is right that the ideal crystal structure with its presented planes is not necessarily the one after milling. The purpose of this figure is not to display crystallographic planes where adsorption has occurred but rather to display the different longitudinal axes – i.e. the c-axis for crystal (a) and the axis a’ for the z-bar (b). The latter is perpendicular to the c-axis. This is essential for understanding the behavior at fracture. Because the first fractures will be (almost) perpendicular to c in (a) but subparallel to c in (b). We have therefore chosen to show almost identical axis orientation in Figure 4. The vertical c-axis is vertical in both cases and parallel to each other, also the a-axes have similar orientations. The only difference is that the crystal in (a) is seen slightly from above (not exactly vertical to c) while the z-bar in (b) is seen exactly perpendicular to the c-axis.
9) Section of conclusion is absent. Please write conclusion of main findings.
Response: Thank you very much for this remark that motivated us to add the following conclusions:
Enantioselective adsorption of alanine and leucine with %ee values ranging from 1% to 55% was observed to occur on homochiral quartz sand. The l-enantiomer of both investigated amino acids was preferably adsorbed regardless of the handedness of the enantiomorphic quartz. We currently explain the lack of mirror symmetry breaking in our experiments with the different crystal habitus of the synthetic z-bar of d-quartz and the natural mountain crystals of l-quartz. Future experiments using d- and l-quartz with similar habitus to minimize random effects are required to conclude on the possible mirror symmetry breaking effect through amino acid adsorption on opposite enantiomorphic minerals.
Despite the absence of %ee values of opposite sign, there are consistent trends in our collected data such as natural l-quartz showed stronger enantioselective adsorption affinities than synthetic d-quartz for both amino acids. Moreover, strong radiation damage clearly reduced the enantioselective adsorption of leucine, while no clear conclusion can be drawn for the results on alanine regarding nonirradiated vs irradiated quartz. Finally, the desorption capacity of alanine and leucine was equally strong in each of the corresponding experiments using individual amino acid solutions as well as using the combined amino acid mixture, i.e. no amino acid selectivity in terms of pure adsorption of alanine vs leucine onto identical quartz surfaces.

Round 2
Reviewer 1 Report
Dear Editor, Dear Authors,
the manuscript
"Racemate resolution of alanine and leucine on homochiral quartz, and its alteration by strong radiation damage" (life-1413494)
got majorly well-revised. However, I still have one, major problem which I think to be crucial to get clarified before publication (in worst case with additional experiments):
Line 366 / Table 3
* Why is there the opposite effect that alanine shows stronger enantiosepation for irradiated d-quartz when probing a mixed ala-leu solution, compared to single compound experiments and leucine in mixture, in which nonirradiated quartz shows stronger enantioseparation?
* Why is the effect different for ala for nonirr. vs irradiated d- and l-quartz? For leu, there is stronger enantioseparation for nonirr. d- and l-quartz while for ala, it is stronger for nonirr quartz only in d form but weaker for the nonirr. l form.
* Why is the effect in enantioseparation stronger for leucine than for alanine (difference in %ee for nonirr. vs irr. quartz for both d- and l-quartz)? Is there a molecular size-related effect?
Unfortunately, the author's responses have been not very helpful yet.
Author Response
I am writing with reference to our Life manuscript life-1413494 entitled “Racemate resolution of alanine and leucine on homochiral quartz, and its alteration by strong radiation damage”. We highly appreciate the time that you have dedicated again to read our revised manuscript and we are grateful for the opportunity to submit a (re-)revised manuscript incorporating as best as possible, and where appropriate, your additional comments. Please find below our point-by-point responses to your comments. Changes in the manuscript have been highlighted in dark grey.
Reviewer 1
Dear Editor, Dear Authors,
the manuscript "Racemate resolution of alanine and leucine on homochiral quartz, and its alteration by strong radiation damage" (life-1413494) got majorly well-revised. However, I still have one, major problem which I think to be crucial to get clarified before publication (in worst case with additional experiments):
Line 366 / Table 3
* Why is there the opposite effect that alanine shows stronger enantiosepation for irradiated d-quartz when probing a mixed ala-leu solution, compared to single compound experiments and leucine in mixture, in which nonirradiated quartz shows stronger enantioseparation?
Response: As already highlighted in our previous response and the manuscript, we cannot exclude any random effects due to different packing densities of the quartz grains in the funnels vs. the thin tubes. In general, the grain packing in the funnels used for the adsorption experiments of the individual amino acid solutions should correspond much better to the textural conditions in a natural sediment exhibiting almost no wall effects. We have addressed these experimental difficulties in the previous revised version (Line 441 ...) and added the following sentence: “The grain packing in the funnels corresponds much better to the textural conditions in natural sediments and will be chosen as preferred experimental design for future adsorption experiments.”
Please also see our statement in Line 451: “Further competitive enantioselective adsorption experiments are required to study the profound impact of intimate binding between enantiomorphic quartz and amino acids of different size, charge, and polarizability, as well as mutual effects between different molecular species during competitive adsorption.“
It is evident to us that our first study on the enantioselective adsorption of amino acids on homochiral quartz sand raises several more questions and we are therefore glad to pursue our experiments on the promising possibility to induce enantiomeric enrichment via minerals surfaces. At the moment, we could only provide further hypotheses to explain our results which we believe would only weaken the conclusions of our manuscript. To fully address the (potential) difference in enantioselective adsorption among different amino acids, we are currently planning a systematic screening of individual amino acids adsorbed under identical conditions onto d- and l-quartz.
Our previous Response: Regarding the experiments with leucine, the radiation damage clearly reduces the enantioselective adsorption capacity of quartz (please see manuscript page 11). We explained this in the text with potential metamictization of the crystal lattice due to radiation-induced destruction of the crystalline structure. The results for leucine are consistent in this regard. Alanine, however, shows inconsistent data in terms of the evolution of ee in the experiments with and without irradiation. At this point we can only speculate, and additional experiments with similar d- and l- quartz samples as discussed in the manuscript are required to conclude on the results of alanine as well as the experiments using amino acid mixtures to understand competitive adsorption effects.
* Why is the effect different for ala for nonirr. vs irradiated d- and l-quartz? For leu, there is stronger enantioseparation for nonirr. d- and l-quartz while for ala, it is stronger for nonirr quartz only in d form but weaker for the nonirr. l form.
Response: The referee is right that the results for leucine seem consistent regarding the effect of radiation damage, which reduces the enantioseparation consistently by a factor of ≥2. However, given the current dataset, we are not willing to introduce additional speculations on the (potentially) inconsistent data for alanine when comparing the effect of the radiation damage on the enantioselective adsorption of alanine with the one of leucine. Both, competitive adsorption effects and molecular size-related effects, can be considered but require additional research with several other amino acids and a better control of grain habitus and packing densities. Our pilot study has shown that both handedness and absolute enantiomeric enrichment are controlled by many factors that we do not yet fully understand. Our first data are very promising and should stimulate further experiments and discussions on the potential role of minerals on the enantioselective resolution of chiral biomolecules.
* Why is the effect in enantioseparation stronger for leucine than for alanine (difference in %ee for nonirr. vs irr. quartz for both d- and l-quartz)? Is there a molecular size-related effect?
Response: Indeed, we can imagine that size- and polarization-related effects can play a role but for a well-sounded conclusion a series of adsorption experiments with other amino acids would be necessary (i.e. amino acids of different size and functional groups). With the database of the present pilot study (two aliphatic amino acids), such assumptions are too speculative. We have indicated this in the text (Line 452). To best account for the reviewer’s comment, we added the following explanations to the discussions (Line 500): “Overall, leucine showed higher racemate resolution on quartz with a maximum %ee of 55.6% compared with alanine in our experiments. Molecular size-related effects may be responsible for the observed trend and future investigations on various individual amino acid solutions covering different molecular sizes / branching as well as different functional groups (aliphatic, aromatic, acidic, basic, etc.) are required to study size- and polarization-related effects on the enantioselective adsorption behavior.”
Our previous Response: We can imagine that size- and polarization-related effects can play a role but based on these very first results such assumptions are too speculative for a well-sounded conclusion. This will be a subject for future investigations that should include amino acids of different size and functional groups (aliphatic, aromatic, acidic, basic, cyclic, etc.) in individual and combined solutions.
